

# Using underwater video to evaluate the performance of the Fukui trap as a mitigation tool for the invasive European green crab (*Carcinus maenas*) in Newfoundland, Canada

Jonathan A. Bergshoeff[1,2], Cynthia H. McKenzie[1,3], Kiley Best[4], Nicola Zargarpour[1,2] and Brett Favaro[1,2]

[1] Department of Ocean Sciences, Memorial University of Newfoundland, St. John's, Newfoundland and Labrador, Canada
[2] Centre for Sustainable Aquatic Resources, Fisheries and Marine Institute of Memorial University of Newfoundland, St. John's, Newfoundland and Labrador, Canada
[3] Northwest Atlantic Fisheries Centre, Fisheries and Oceans Canada, St. John's, Newfoundland and Labrador, Canada
[4] Centre for Fisheries and Ecosystems Research, Fisheries and Marine Institute of Memorial University of Newfoundland, St. John's, Newfoundland and Labrador, Canada

Corresponding author
Jonathan A. Bergshoeff,
jon.bergshoeff@gmail.com

## ABSTRACT

The European green crab (*Carcinus maenas*) is a destructive marine invader that was first discovered in Newfoundland waters in 2007 and has since become established in nearshore ecosystems on the south and west coast of the island. Targeted fishing programs aimed at removing green crabs from invaded Newfoundland ecosystems use Fukui traps, but the capture efficiency of these traps has not been previously assessed. We assessed Fukui traps using *in situ* observation with underwater video cameras as they actively fished for green crabs. From these videos, we recorded the number of green crabs that approached the trap, the outcome of each entry attempt (success or failure), and the number of exits from the trap. Across eight videos, we observed 1,226 green crab entry attempts, with only a 16% rate of success from these attempts. Based on these observations we believe there is scope to improve the performance of the Fukui trap through modifications in order to achieve a higher catch per unit effort (CPUE), maximizing trap usage for mitigation. Ultimately, a more efficient Fukui trap will help to control green crab populations in order to preserve the function and integrity of ecosystems invaded by the green crab.

# INTRODUCTION

The European green crab, *Carcinus maenas* (Linnaeus, 1758) is a crustacean species native to European and North African coastlines (*Williams, 1984*). It has been ranked among 100 of the world's 'worst invasive alien species' by the International Union for Conservation of Nature (*Lowe et al., 2000*). In North America, current distributions of the European

green crab (hereafter green crab) on the west coast range from California, USA (*Cohen, Carlton & Fountain, 1995*; *Yamada et al., 2008*) up to British Columbia, Canada (*Gillespie et al., 2007*). On the east coast green crabs can be found from Virginia, USA (*Williams, 1984*) to Newfoundland, Canada (*Blakeslee et al., 2010*; *McKenzie et al., 2011*). Evidence suggests that green crab populations on the east coast are made up of both northern and southern genotypes that originated from two separate introduction events. First, the historical invasion of the northeastern United States in the early 1800's by green crabs originating from the southern UK (*Say, 1817*; *Roman, 2006*; *Blakeslee et al., 2010*). Second, an introduction into the Maritimes in the late 1980's by a more cold-tolerant population from the northern limit of the green crab's range in Europe (*Roman, 2006*; *Blakeslee et al., 2010*; *DFO, 2011a*).

The green crab was first discovered in the nearshore waters of Newfoundland in 2007, and has since become established across the southern and western coasts of the island (*DFO, 2011a*). Genetic analysis of green crab populations indicate a mixed ancestry of both the southern and northern genotypes, with a close relationship to the more cold-tolerant, northern population (*Blakeslee et al., 2010*; *DFO, 2011a*). Recent findings show that green crab populations on the west coast of Newfoundland (i.e., St. George's Bay) are genetically different from those on the southeast coast (i.e., Placentia Bay), which could manifest itself in different behaviours and invasion characteristics (*Rossong et al., 2012*; *Jeffery et al., 2017*). The invasion is concerning because green crabs destroy eelgrass beds (*DFO, 2011a*; *Matheson et al., 2016*), are voracious predators of bivalves (*Ropes, 1968*; *Cohen, Carlton & Fountain, 1995*; *Klassen & Locke, 2007*; *Matheson & McKenzie, 2014*), and compete with native species and other crustaceans for food and habitat (*Cohen, Carlton & Fountain, 1995*; *Matheson & Gagnon, 2012*). The impact of green crabs on eelgrass beds is particularly threatening as invasive species are one of the multiple stressors contributing to a global trend in seagrass decline (*Orth et al., 2006*). Eelgrass serves as important habitat for commercial species such as cod, herring, and lobster. Therefore, green crab invasions pose both an ecological and economic threat (*Joseph, Schmidt & Gregory, 2013*; *Matheson et al., 2016*).

The complete eradication of an invasive species in an aquatic environment is virtually impossible once the organism has become established, unless the invasion is in a confined area and addressed shortly after arrival (*Bax et al., 2003*; *Lodge et al., 2006*). In Newfoundland, the complete eradication of green crabs is no longer considered an option. Therefore, efforts are now focused on mitigation to suppress invasive populations to slow their spread and minimize their negative effects (*DFO, 2011b*). These mitigation studies have found that the direct removal of green crabs through focused trapping is one effective control technique, and has become the current method of conducting targeted removals of green crabs on both the east and west coast of Canada (*DFO, 2011a*; *DFO, 2011b*; *Duncombe & Therriault, 2017*). Green crab removal efforts in Canada usually utilize Fukui traps (60 × 45 × 20 cm, 12 mm bar length square mesh, 45 cm expandable entry slit) which are practical for mitigation efforts as they are light-weight, collapsible, durable, and can be easily deployed from small boats or from shore.

Despite the widespread use of the Fukui trap for research, monitoring, and mitigation, there have been no formal investigations of the interactions between green crabs and the

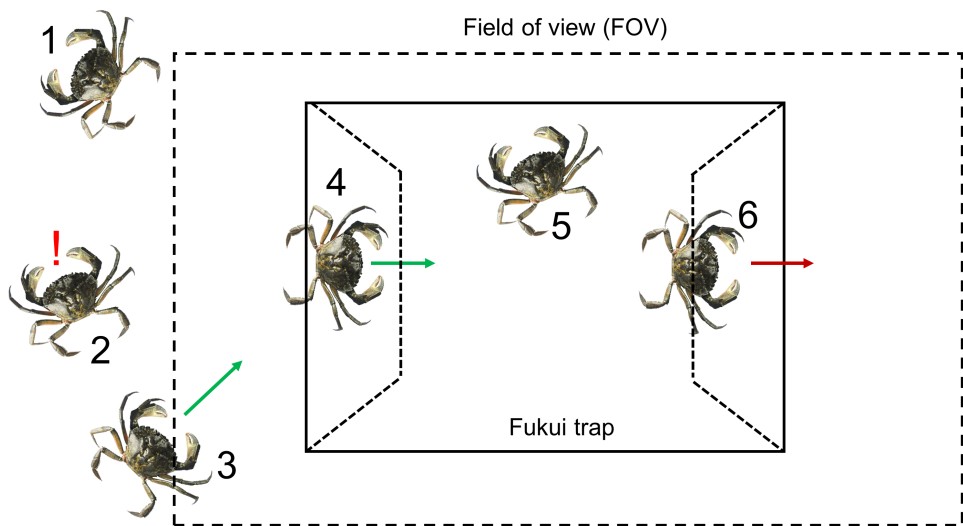

**Figure 1** **A visual representation of the six steps required for a green crab to be captured.** The numbers indicate the step in the capture process: (1) Presence, (2) Detection, (3) Approach, (4) Attempt, (5) Capture, (6) Exit.

standard Fukui trap, and substantial knowledge gaps exist surrounding the trap's overall efficiency. In addition, it has been shown that green crab aggression and feeding behaviour can vary across sites, which may influence catch rates and the performance of the trap between areas (*Rossong et al., 2012*). The main objectives of this study were to evaluate the performance and efficiency of the Fukui trap in terms of its ability to catch green crabs, and to gain a better understanding of this capture process and how it may differ across sites in Newfoundland.

In this study, we used underwater video cameras to record footage of the traps as they actively fished for green crabs *in situ* across Newfoundland. Underwater video is the best way to understand the interactions between an animal and a piece of fishing gear, and is beneficial in determining the optimal design and use of this fishing gear (*Favaro et al., 2012*; *Underwood, Winger & Legge, 2012*). There is a growing body of literature on the use of cameras to better understand various types of fishing gears, including traps (alternatively referred to as pots) (*Jury et al., 2001*; *Barber & Cobb, 2009*; *Bacheler et al., 2013*; *Favaro, Duff & Côté, 2013*; *Meintzer, Walsh & Favaro, 2017*), trawls (*Nguyen et al., 2014*; *Underwood et al., 2015*), and hooks (*He, 2003*; *Robbins et al., 2013*). In the case of the Fukui trap, underwater video is an effective method to accurately assess the number of green crabs that approach the trap, the outcome of each attempt to enter the trap, and the likelihood that a green crab will remain inside the trap before it is retrieved.

Six steps have to be completed successfully for green crabs to be caught in a trap (Fig. 1) (*Favaro, Duff & Côté, 2014*). First, they must be present in the area where the Fukui trap has been deployed. Second, they must be able to detect the presence of the trap, either visually or by detecting olfactory cues of the bait plume. Third, green crabs must approach the Fukui trap. Fourth, they must locate one of the entrances and make an entry attempt.

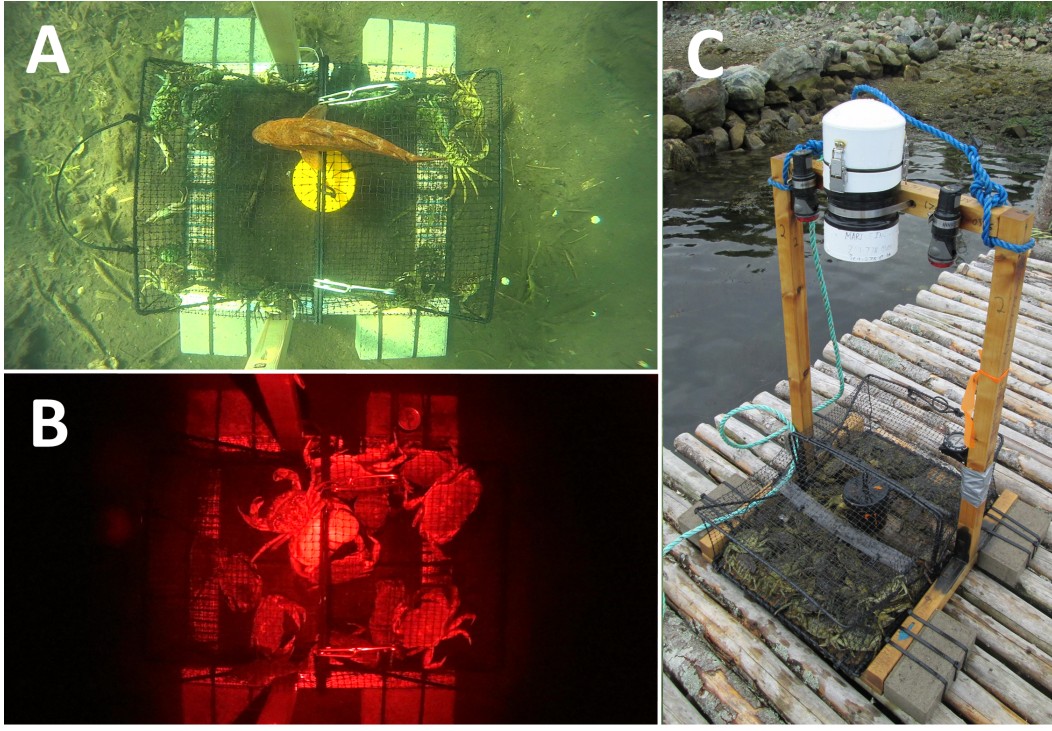

**Figure 2** **The camera frame constructed around a Fukui trap and its field of view.** A top-down view of the Fukui trap recorded during a daytime deployment (A) and overnight deployment (B). The entire camera apparatus mounted to a Fukui trap (C).

Fifth, they must successfully complete that entry attempt in order to become captured. Sixth, they must remain in the trap until the gear is hauled (i.e., they must not exit). The use of underwater video cameras in this study enabled us to accurately evaluate steps three through six of the capture process (number of approaches to the trap, proportion of successful entry attempts, number of exits) in order to determine the effectiveness of the Fukui trap at catching green crabs. Furthermore, the use of underwater video allowed us to identify barriers that were inhibiting the capture process. This information will enable us to identify inefficiencies in the capture process that could be addressed through modifications to the fishing gear, so that future removal programs can be conducted more efficiently.

## METHODS

### Camera apparatus and equipment

We used custom-built camera housings with Sony HDR-AS20 Action Cameras capable of recording 13-hour high-definition underwater videos (as described in *Bergshoeff et al., 2017*). We mounted each camera system to a wooden frame built around a standard Fukui trap. Using a large 114–165 mm diameter gear-clamp, the camera housing was centred above the trap, with the camera pointing downward to provide a top-down view of the trap and surrounding area (Fig. 2). The camera was positioned at a height of 53 cm above the top of the trap and 74 cm above the ocean floor, creating a field-of-view (FOV) of
approximately 81 cm by 150 cm when filming underwater. The wide-angle lens of the camera made it possible to view the entire trap, in addition to a buffer surrounding all edges of the trap (45 cm to the left and right edge of the trap, and 18 cm from the top and bottom edge). The wooden frame was weighted down with four 2.8 kg cement bricks in order to make it negatively buoyant and to prevent shifting due to currents and wave action. Finally, the rope attaching the trap to the surface float was marked in half-metre increments in order to determine the approximate depth of deployment.

An external lighting system was necessary for overnight trap deployments; therefore, each camera apparatus was equipped with two Light and Motion (Marina, California, USA) GoBe Plus flashlights with red LED light attachments (GoBe Focus Head). On low-power mode these flashlights had sufficient battery life to illuminate the entire night cycle. Many crustaceans are insensitive to wavelengths greater than 620 nm; therefore, we used red lights with the goal of minimizing the behavioural impacts that may accompany full-spectrum light (*Nguyen et al., 2017*).

## Field methods

We recorded underwater videos at six sites across Newfoundland during the summer of 2015 and one site during the summer of 2016 (Fig. 3). We produced the map in Fig. 3 using the ggmap package (*Kahle & Wickham, 2013*) in R (*R Core Team, 2015*). The sites were as follows: 1. Fair Haven (FH), Placentia Bay (June 9–11, 2015 and August 18–20, 2015) 2. Boat Harbour (BH), Placentia Bay (June 23–26, 2015) 3. Little Harbour East (LHE), Fortune Bay (June 22–23, 2015) 4. Little Port Harmon (PH), St. George's Bay (July 7–10, 2015) 5. Penguin Arm (PA), Bay of Islands (July 14–15, 2015) 6. Deer Arm (BB), Bonne Bay (July 11–14, 2015) 7. Fox Harbour (FX), Placentia Bay (June 30–July 1, 2016). Each of these shallow, coastal sites have known green crab populations, and consist of similar mixed mud, sand, and rock habitat. The video data from June 2016 in Fox Harbour, NL were collected as part of a complementary study that followed the same methodology for recording videos, and we therefore included the results in our analysis.

At each site we followed a set procedure for deploying the camera traps. Prior to each deployment, the Fukui traps were baited with equal amounts of Atlantic herring (*Clupea harengus*), the standard bait used by Fisheries and Oceans Canada (hereafter, DFO) for green crab mitigation projects, in a perforated plastic bait container (*Gillespie et al., 2007*; *DFO, 2011b*). The herring was thawed and cut into pieces, with approximately half of a fish placed into each bait container. Once the traps were baited, the camera equipment was secured inside the camera housing and mounted to the frame surrounding the Fukui trap. We used a wireless Sony RM-LVR1 Live View Remote to ensure that the camera and FOV were oriented correctly and to initiate recording prior to each trap deployment.

We typically deployed the traps close to shore (<50 m) using a small Zodiac boat. When we placed the traps in the water, we made sure that the camera housing entered the water horizontally in order to prevent air bubbles from becoming trapped on the housing's acrylic viewport. We deployed each trap no less than 1 m below the low tide water depth to prevent the camera apparatus from breaching the surface with the changing tides. Each camera trap was paired with a Fukui trap without an attached camera to examine whether the camera

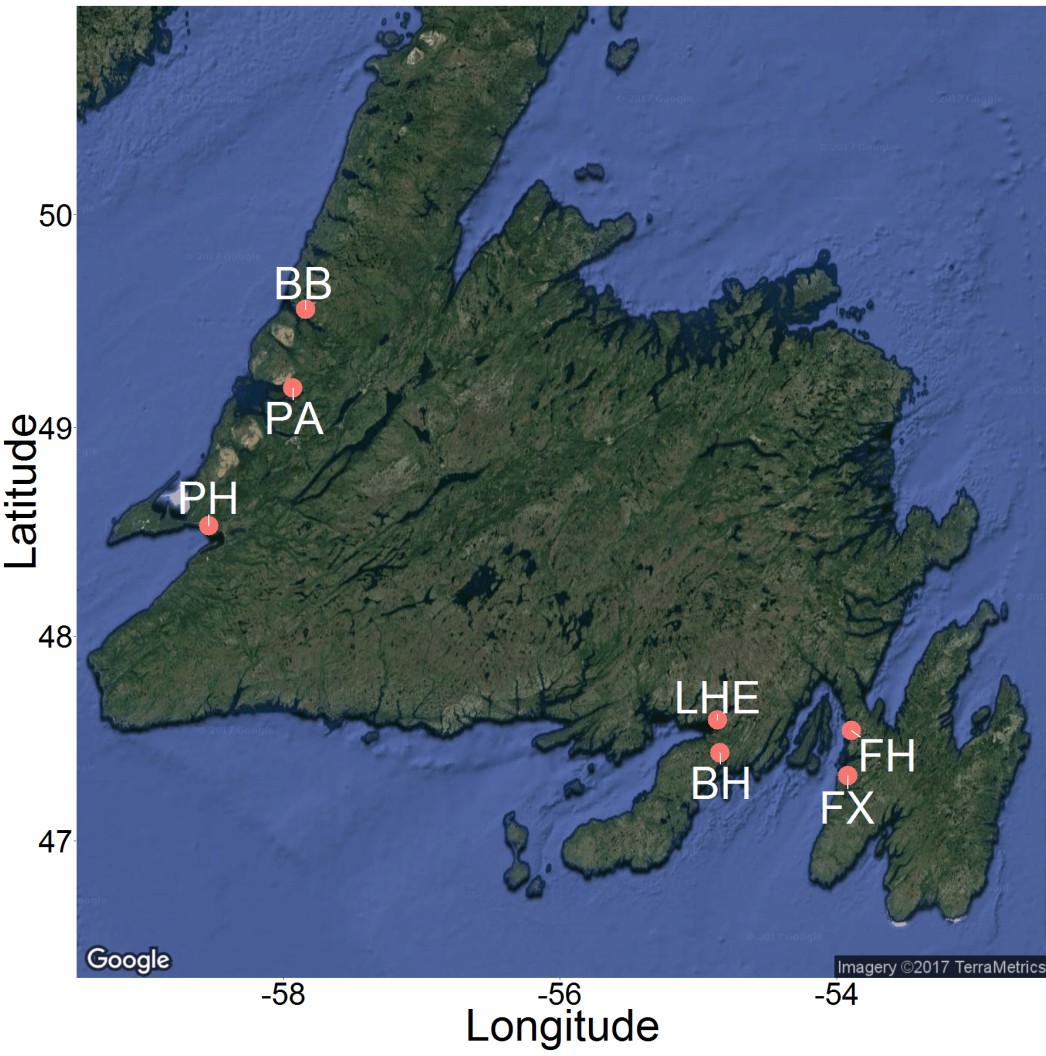

**Figure 3** **Map of 2015 and 2016 study sites across Newfoundland.** Sites included Bonne Bay (BB), Boat Harbour (BH), Fair Haven (FH), Little Harbour East (LHE), Penguin Arm (PA), Little Port Harmon (PH), and Fox Harbour (FX). Map imagery ©2017 TerraMetrics.

itself affected catch rates. The two traps within each pair were placed approximately 10 m apart based on other studies involving the Fukui traps (*Gillespie et al., 2007*; *Yamada et al., 2008*; *Curtis et al., 2015*). In total, two camera traps and two non-camera traps were set at each deployment. Sampling location, global positioning system (GPS) coordinates, time of day, depth, and weather information were recorded for each deployment. Traps were either deployed early in the day and retrieved in the evening (termed 'daytime deployments'), or deployed before sunset and retrieved the next morning (termed 'overnight deployments'). We aimed for each trap to be deployed for 12 h, but logistical factors such as weather and travel sometimes affected trap retrieval time adding variation to total soak time. These logistical factors also meant that some traps were not deployed until the afternoon, and retrieved the following morning (termed 'mixed deployments').

**Table 1  Summary of green crabs caught at each study site in 2015.**

| Location | Deployments (n) | Mean catch | Standard deviation | Minimum catch | Maximum catch | Total catch |
|---|---|---|---|---|---|---|
| Fair Haven | 16 | 131.8 | 88.2 | 10 | 299 | 2,108 |
| Little Port Harmon | 12 | 34.4 | 30.4 | 0 | 102 | 413 |
| Little Harbour East | 4 | 0.5 | 1 | 0 | 2 | 2 |
| Penguin Arm | 4 | 0 | 0 | 0 | 0 | 0 |
| Bonne Bay | 20 | 0.3 | 0.6 | 0 | 2 | 5 |
| Boat Harbour | 22 | 8.6 | 34.8 | 0 | 164 | 188 |
| All Sites | 78 | 34.8 | 67.5 | 0 | 299 | 2,716 |

When the traps were retrieved the catch was sorted, counted, and sexed. All bycatch species were visually identified to the lowest possible taxonomic level, recorded and released as soon as possible. As per DFO recommendations, all captured green crabs were euthanized by freezing and disposed of. Once the catch was processed, the camera equipment was reset, and the traps were prepared for re-deployment. We re-baited the traps with fresh herring before each new deployment.

The project was approved as a 'Category A' study by the Institutional Animal Care Committee at Memorial University as it involved only invertebrates (project # 15-02-BF), and all field research was conducted under experimental licenses NL-3133-15 and NL-3271-16 issued by DFO.

Throughout our manuscript, data description was done using the mean and standard deviation (SD). When reporting a mean, we included the SD in parentheses. When reporting a range, we included the mean and SD in parentheses.

## Determining the effect of camera presence on catch

We built two linear mixed-effects models using the nlme package (*Pinhero et al., 2017*) in R (*R Core Team, 2015*) in order to test whether the presence of the camera had an effect on green crab catch. We analyzed a subset of the green crab catch data which included only Fair Haven, NL and Little Port Harmon, NL. All other sites were excluded from our subset due to either zero green crab catch, or low mean catch rates (Table 1). We did not see any meaningful relationship between deployment duration and catch (Figs. 4A and 4B). However, the soak times were not consistent between Fair Haven (range = 21.8–24.3 h; mean = 22.9 h; SD = 0.8) and Little Port Harmon (range = 7.4–14.1 h, mean = 11.1 h; SD = 2.8) (Fig. 4C). To account for this, we created a separate model for each location because the underlying effect of soak time on catch was potentially unique to each site (Fig. 4D). These two models tested the fixed effects of camera presence (i.e., camera present, camera absent) and duration on catch-per-deployment. We did not include deployment type (i.e., daytime, overnight, mixed) in our final models as the term was insignificant in the Little Port Harmon model, and lacked sufficient factor levels in the Fair Haven model (mixed deployments only). Due to the paired nature of our design we designated each camera and non-camera pair as a single deployment, which was included in each model as a random effect. The residuals for both the Fair Haven and Little Port Harmon models met the assumptions for homogeneity, normality, and independence.

 

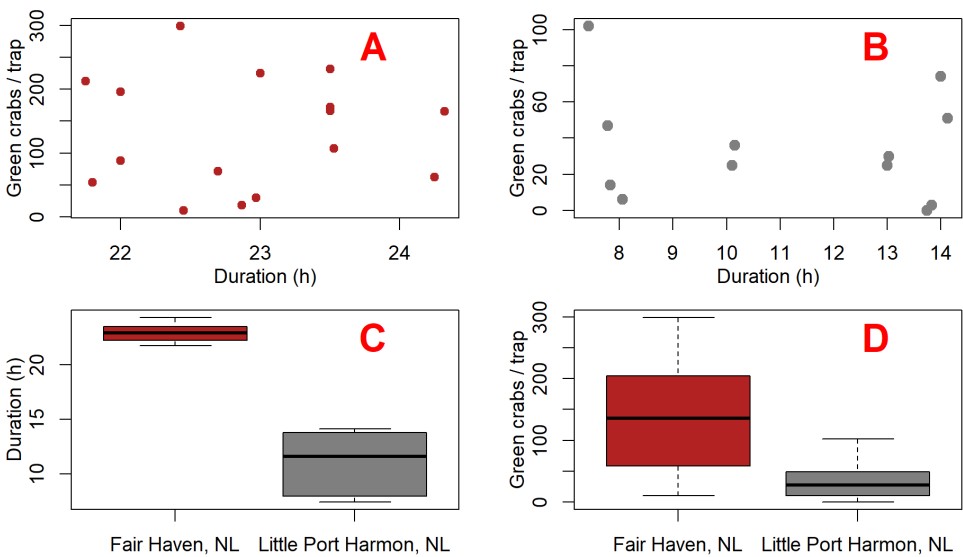

**Figure 4** **Plots comparing green crab catch and fishing duration between Fair Haven, NL and Little Port Harmon, NL.** Scatterplots (A and B) show the duration and number of green crabs captured for each deployment at Fair Haven (A) ($n = 16$) and Little Port Harmon (B) ($n = 13$), respectively. Boxplot (C) shows the mean deployment duration at Fair Haven and Little Port Harmon. Boxplot (D) shows the mean green crab catch per Fukui trap at Fair Haven and Little Port Harmon.

## Video analysis

### Video selection

In order to determine which videos to analyze in-full, we first reviewed them according to a selection key (Fig. S1). This process involved evaluating the level of green crab activity in each video, as well as an assessment of the overall image quality. The activity level of each video was determined by counting the approximate number of green crabs present in the field of view (FOV) at 35-min intervals and calculating the overall mean across those intervals. The average number of green crabs in the FOV corresponded to the following activity levels: 0 = 'none'; 0.1–5.0 = 'low'; 5.1–10.0 = 'medium'; 10.0 and above = 'high'. If the activity level was determined to be 'none' or 'low' the video was disqualified. Our assessment of video quality was based on visibility of the trap due to particulate matter and lighting conditions. If the lower panel (i.e., the floor) of the Fukui trap was clearly visible, as well as the entire periphery of the FOV, then the video quality was classified as 'good'. If the lower panel of the Fukui trap was clearly visible, but the periphery of the FOV was poorly lit, then the video quality was classified as 'fair'. Finally, if the lower panel of the Fukui trap was not visible due to lighting or particulate matter, the video quality was classified as 'poor'. If the video quality was determined to be 'poor', the video was disqualified. Overall, in order to qualify for analysis each video required 'medium' or 'high' activity levels, as well as 'fair' or 'good' video quality.

### Video analysis procedure

We used a standardized procedure to evaluate the video obtained during the 2015 and 2016 field seasons. Video files were viewed using VLC Media Player 2.2.4 on a 27-inch (68.6 cm)

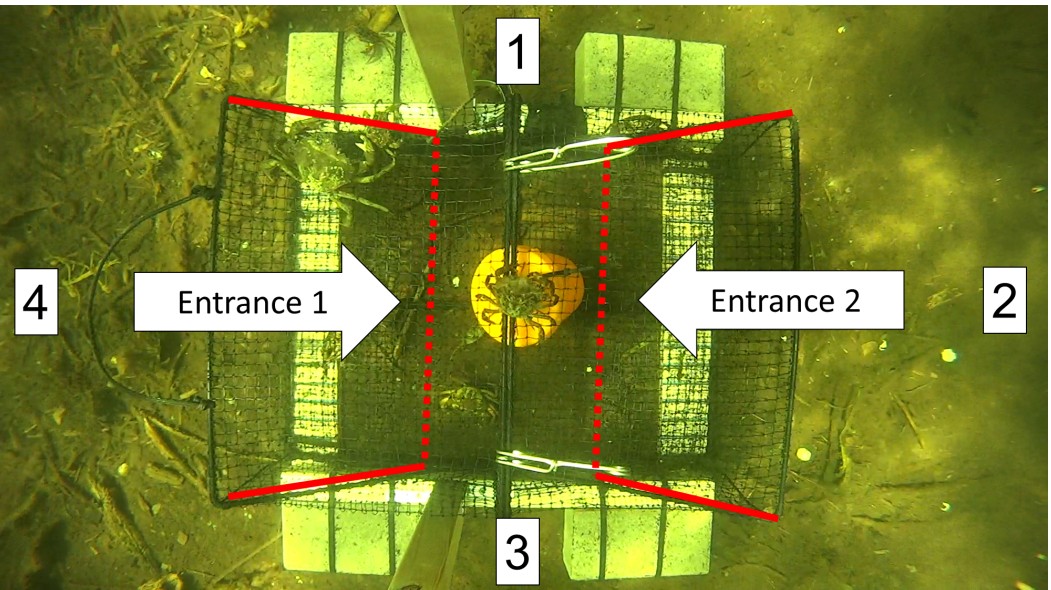

**Figure 5 A screen shot from a video recording showing the top-down view of a Fukui trap as it actively fishes *in situ*.** Approaches were recorded every time an animal entered the FOV from direction 1, 2, 3, or 4. The entrance tunnels are outlined with red lines. The dotted red line indicates the entry slits into the trap.

16:9 (widescreen) flat screen monitor. For night videos, we used the sepia colour setting in VLC to reduce glare and eye-strain caused by the red lighting. Data were recorded in a spreadsheet using Microsoft Excel 2013. The analysis procedure involved characterizing the video by 'events' (both qualitative and quantitative) and recording the time during the video at which each event occurred.

We began analyzing the video as soon as the trap settled on the ocean floor after deployment. The FOV was divided into four sections in a clockwise manner (top = 1, right = 2, bottom = 3, left = 4). Every time an animal entered the FOV, we recorded the direction of approach (e.g., APP1, APP2), the species (e.g., GC for green crab, RC for rock crab), and the time as indicated by the VLC time counter. A rough estimate of size was made for each species (small, medium, or large); however, limited emphasis was placed on this information due to the potential for biases and size distortion depending on the distance of a green crab from the camera.

We recorded each attempt to enter the trap, along with the time taken to complete or fail the attempt. For green crabs, an attempt was defined as when the entire body of the crab was inside the entry tunnel of either entrance 1 or entrance 2 (Fig. 5). The time for each attempt was recorded until the entry was either successful (i.e., a green crab fully entered the trap) or failed (i.e., a green crab fully left the entrance tunnel). If an entry attempt failed, the predominant reason for failure was noted according to four common, reoccurring situations: (1) Agonism (AGON): some form of intraspecific or interspecific agonistic behaviour deterred or prevented the green crab from entering the trap, (2) Partial entry (PE): the green crab entered the entrance tunnel, but turned around and exited before

contacting the trap entry slit, (3) Full entry (FE): the green crab fully entered the entrance tunnel and contacted the trap entry slit, but subsequently turned around and exited, or (4) Difficulty completing entry (DCE): the green crab fully entered the entrance tunnel, but was unable to get through the trap entry slit in order to successfully complete the entry, and subsequently turned around and exited. Additionally, if a green crab was able to escape the trap after it had successfully entered, this was recorded as an exit.

If a notable behaviour occurred that was not part of our core observation framework (e.g., predation) we recorded the time and context of the event. We focused on behavioural interactions outside of the trap instead of green crabs already inside the trap, which could be seen as an artificial environment influencing behaviour.

### Regional performance of the Fukui trap

Recently, it has been shown that genetically different green crab populations exist within Newfoundland which could influence behaviour and catchability (*Rossong et al., 2012*; *Jeffery et al., 2017*). We compared video analysis results between St. George's Bay (i.e., Little Port Harmon) on the west coast of Newfoundland, and Placentia Bay (i.e., Fair Haven and Fox Harbour) on the southeast coast in order to examine regional differences in the performance of the Fukui trap. When comparing these regional differences, we focused on parameters related directly to the interactions of green crabs with the Fukui trap. This allowed us to evaluate whether variations in regional green crab behaviour had an impact on Fukui trap performance. The two parameters we examined were the elapsed time for successful and failed entry attempts, and the frequency of these attempts.

We used a generalized linear mixed model (GLMM) to test whether there was an interaction between the elapsed time for successful or failed green crab entry attempts, and region. To build our model we used the lme4 package (*Bates et al., 2017*) in R (*R Core Team, 2015*). The distribution of elapsed entry attempt time was best explained by a negative binomial distribution. The fixed covariates in our model were *outcome* (categorical with two levels: success, failure) and *region* (categorical with two levels: west, southeast). We included *video ID* as a random effect to account for dependency among observations from the same video. We verified the assumptions of our model by plotting residuals versus fitted values, and testing for overdispersion.

We assessed whether there was an association between the frequency of entry attempt outcomes (i.e., successful entry, failed entry) and region (i.e., west, southeast) using a chi-squared test. We set the level of statistical significance for rejecting the null hypothesis at $p < 0.05$.

## RESULTS

### Field deployments

During the 2015 field season, a total of 39 camera traps and 39 traps without cameras were deployed (total $n = 78$) across the six field sites. Trap deployment times ranged from 2.7 to 24.4 h (mean = 14.2 h; SD = 6.1). We collected 37 videos in total (Table S1). Two of the 39 videos failed due to partial flooding of the camera housing. Recording duration of videos ranged from 2.7 to 13.0 h (mean = 11.2 h; SD = 2.7). The inconsistency in deployment

**Table 2 Summary of all bycatch species caught at each study site in 2015.** The number of green crabs caught at each site has also been included for comparison purposes.

| | Fair Haven | Little Port Harmon | Boat Harbour | Little Harbour East | Bonne Bay | Penguin Arm | All sites |
|---|---|---|---|---|---|---|---|
| Rock crab (*Cancer irroratus*) | 0 | 2 | 116 | 0 | 84 | 4 | 206 |
| Cunner (*Tautogolabrus adspersus*) | 0 | 0 | 8 | 0 | 39 | 7 | 54 |
| Winter flounder (*Pseudopleuronectes americanus*) | 1 | 2 | 3 | 0 | 1 | 2 | 9 |
| Sculpin sp. (*Myoxocephalus sp.*) | 0 | 0 | 0 | 0 | 2 | 2 | 4 |
| American eel (*Anguilla rostrata*) | 0 | 1 | 0 | 0 | 0 | 1 | 2 |
| Green crab (*Carcinus maenas*) | 2,108 | 413 | 188 | 2 | 5 | 0 | 2,716 |

durations can be attributed to a combination of logistical challenges getting to-and-from the site and inclement weather preventing retrieval of the gear.

Both the fishing effort and the number of green crabs caught per trap varied across the six study sites visited in 2015, with all but two of the sites (Fair Haven and Little Port Harmon) exhibiting a mean catch of less than 10 green crabs per deployment (Table 1). Generally, bycatch using the Fukui trap was minimal. The most common occurrence of bycatch was rock crab (*Cancer irroratus*) in Boat Harbour and Bonne Bay (Table 2).

## Camera effects

We found the presence of the camera had no significant impact on catch at both Fair Haven ($\beta 1 = 19.409$, S.E. $= 46.797$, $t = 0.415$, $p = 0.693$) and Little Port Harmon ($\beta 1 = -15.951$, S.E. $= 16.970$, $t = -1.268$, $p = 0.273$) based on our subset of catch data from these two locations. The effect size, $\beta 1$, can be interpreted as an increase of 19 crabs per trap when a camera is present at Fair Haven, and a decrease in 16 crabs per trap when the camera is present at Little Port Harmon, both relative to non-camera traps. Camera traps fished in Fair Haven ($n = 8$) caught between 10 and 299 green crabs (mean $= 140.9$ crabs; SD $= 99.6$), and non-camera traps fished in Fair Haven ($n = 8$) caught between 18 and 232 green crabs (mean $= 122.6$ crabs; SD $= 80.9$). Camera traps fished in Little Port Harmon ($n = 6$) caught between three and 74 green crabs (mean $= 26.3$ crabs; SD $= 26.3$), and non-camera traps ($n = 6$) fished in Little Port Harmon caught between 0 and 102 green crabs (mean $= 42.5$ crabs; SD $= 34.4$).

## Video analysis

Using the video selection key (Fig. S1), we determined that 8 of the 37 collected videos were suitable for complete analysis (Table S2). The majority of videos that were rejected from the analysis process showed no or 'low' green crab activity. Overall, videos were clear and well illuminated. However, videos collected at night under red illumination were dim around the periphery of the FOV (Fig. 2). Additionally, videos collected in Fair Haven in late-August, 2015 were disqualified due to 'poor' quality caused by excessive turbidity and suspended particulate material in the shallow bay in which we were trapping.

Results from the eight videos that were analyzed can be examined in Table 3. The variability among videos, and the range of green crab activity levels across each site are illustrated in Fig. 6. In total, we observed 2,373 green crab approaches to the trap over the
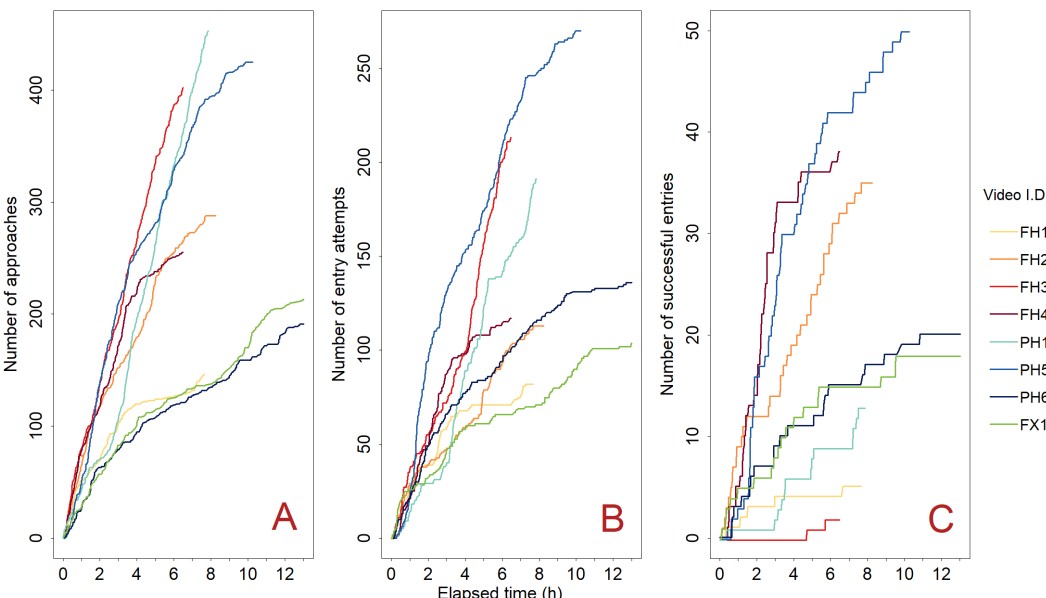

**Figure 6 Green crab accumulation over the course of each trap deployment ($n = 8$).** Green crab approaches (A), entry attempts (B), and accumulation in the Fukui trap (C). We did not observe any exits; therefore, (C) represents both the number of successful entries, and the number of green crabs in the trap. Each coloured line (Video ID) represents the individual deployment of a camera-equipped Fukui trap at either Fair Haven (FH) (red-orange colour scheme), Little Port Harmon (PH) (blue colour scheme), or Fox Harbour (FX) (green colour scheme).

**Table 3 Summary of data from each video that was analyzed.** Video code represents the individual deployment of a camera-equipped Fukui trap at either Fair Haven (FH), Little Port Harmon (PH), or Fox Harbour (FX).

| Video code | Date (MM-DD-YY) | Video duration (h) | Green crab approaches (#) | Green crab attempts (#) | Successful entries by green crab (#) | Exits by green crab (#) | Green crab success rate (%) | Approaches by other species (#) | Entry attempts by other species (#) | Successful entries by other species (#) |
|---|---|---|---|---|---|---|---|---|---|---|
| FH1 | 06/09/15 | 7.7 | 146 | 82 | 5 | 0 | 6.1 | 46 | 15 | 1 |
| FH2 | 06/09/15 | 8.3 | 288 | 113 | 35 | 0 | 31.0 | 57 | 0 | 0 |
| FH3 | 06/10/15 | 6.5 | 402 | 213 | 2 | 0 | 0.9 | 83 | 6 | 0 |
| FH4 | 06/10/15 | 6.5 | 255 | 117 | 38 | 0 | 32.5 | 48 | 5 | 1 |
| PH1 | 07/07/15 | 7.9 | 453 | 191 | 13 | 0 | 6.8 | 16 | 0 | 0 |
| PH5 | 07/09/15 | 10.3 | 425 | 270 | 50 | 0 | 18.5 | 16 | 1 | 1 |
| PH6 | 07/09/15 | 13.0 | 191 | 136 | 20 | 0 | 14.7 | 81 | 2 | 0 |
| FX1 | 06/30/16 | 13.0 | 213 | 104 | 18 | 0 | 17.3 | 4 | 1 | 0 |

course of eight videos (73.0 h), and 351 by other species (Fig. 6A). During these videos, green crabs comprised 86.0% (SD = 10.3) of all approaches to the trap, and it took 3.5 min (SD = 3.4) on average for the first green crab to approach the trap (range: 0.9–11.1 min). We observed an average of 35.7 green crab approaches per hour (SD = 18.2) across all eight videos. Only 8.1% (SD = 5.2) of the 2,373 green crab approaches resulted in a successful entry into the Fukui trap. No green crab exits were observed.

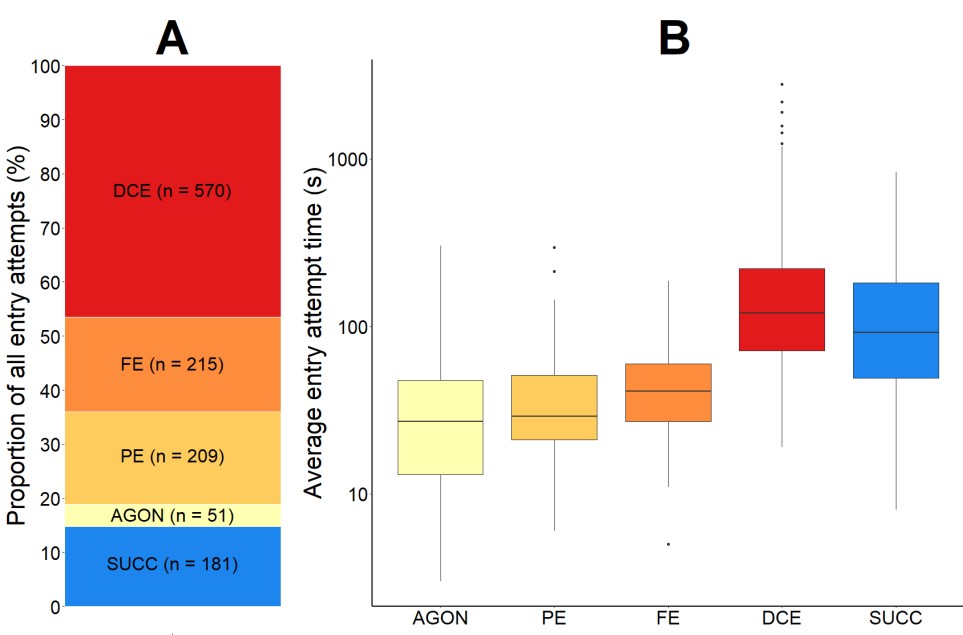

**Figure 7** **The proportional outcome and average time taken for all green crab entry attempts into the Fukui trap.** The proportion of all green crab entry attempts that were successful and that were failures (A). The failed proportion is subdivided according to the four most common reasons for failure: agonistic behaviour (AGON), partial entry (PE), full entry (FE), and difficulty completing entry (DCE). The total number ($n$) of entry attempts for each is given. A boxplot illustrating the average time (seconds, log scale used) for each type of entry attempt (B). The solid black line within the box depicts the median. The lower and upper hinges of the box correspond to the first and third quartiles, respectively. The upper whisker extends to the largest value no further than 1.5 times the inter-quartile range (1.5*IQR). The lower whisker extends to the smallest value no further than 1.5*IQR. Any data points beyond these whiskers are considered outliers, and are plotted individually.

We observed a total of 1,226 green crabs make attempts to enter the Fukui trap across all sites (Fig. 6B), as well as 30 attempts by other species. On average, there were 18.0 entry attempts per hour (SD = 8.9), and 52.5% (SD = 10.9) of the green crabs that approached the trap made entry attempts. In total, 181 green crabs made successful entry attempts (Fig. 6C). The success rate for each video ranged from 0.9–32.5%, with a mean of 16.0% (SD = 11.4). On average, it took a green crab 140.3 s (SD = 147.8) to successfully enter the Fukui trap during an entry attempt (range: 8–837 s), while it took an average of 126.1 s (SD = 200.2) before a green crab would fail an entry attempt (range: 3–2,789 s).

We observed 1,045 failed entry attempts in total. For each of the eight videos, the average proportion of failed entry attempts was 84.0% (SD = 11.4). This proportion can be further broken down according to the four most common reasons for failure (Fig. 7).

First, 4.0% (SD = 2.4) of all entry attempts failed due to some sort of agonistic behaviour (AGON; $n = 51$) preventing the green crab from entering the trap. If two green crabs were making a simultaneous entry attempt, agonistic behaviour between them would often cause either crab to abandon the entry attempt. We also observed crabs already inside of the entry tunnel deterring other crabs from entering. This agonistic behaviour was not

limited to crabs outside of the trap; green crabs that were already successfully captured would occasionally attempt to deter other crabs from entering the trap.

Second, 20.0% (SD = 12.2) of all entry attempts failed because the green crab entered the entry tunnel, but only made a partial entry (PE; $n = 209$) before exiting. There was often no obvious behaviour driving partial entry attempts.

Third, 15.5% (SD = 7.4) of all entry attempts failed after the green crab fully entered the entrance tunnel and contacted the entry slit, but subsequently turned around (FE; $n = 215$). As green crabs moved further inside the wedge-shape entry tunnel towards the entry slit, their movement would become more restricted. Occasionally, the pereopod of a green crab would hook the mesh (1 cm ×1 cm) on the top or side panel of the entrance tunnel, causing the crab to become redirected outside of the trap, instead of further inside.

Finally, 44.5% (SD = 14.4) of all entry attempts failed because the green crab had difficulty getting through the trap entry slit in order to complete the entry (DCE; $n = 570$). The amount of time spent by a green crab attempting to pass through the trap entry slit ranged from 19 to 2,789 s (mean = 194.5 s; SD = 249.3). The sharp pereopods of green crabs would often become entangled or caught in the mesh of the Fukui trap, inhibiting successful entry. Similarly, the five anterolateral spines on either side of the green crabs' eyes would often catch on the mesh of the entry slit during entry attempts. Furthermore, even without getting caught in the mesh of the trap, the entry slit was often too tight for the crabs to easily slip through, causing them to become stuck or entangled, and ultimately fail the entry attempt. If a crab was able to reach one of its pereopods or chelipeds through the trap entry slit, there was often nothing to grab hold of in order to pull itself through the tight-fitting entry slit, resulting in a failed entry attempt.

Based on the 181 successful entry attempts, we observed several scenarios that assisted green crabs in making a successful entry. If a crab approached the entrance tunnel at a fast pace, it was often able to use this momentum to push through the restrictive trap entry slit with minimal effort. Similarly, if a green crab approached the entry slit backwards, this would prevent the forward-facing anterolateral spines of the carapace from becoming caught in the mesh. This would allow green crabs to enter the trap more easily. In other situations, crabs would struggle for sustained periods of time to pass through the trap entry slit, with some eventually achieving success. We also observed crabs using the bait container hanging in the centre of the trap to assist in pulling themselves through the entry slit.

### Regional performance of the Fukui trap

The performance of the Fukui trap remained consistent across Newfoundland, regardless of region (i.e., Fox Harbour and Fair Haven in southeastern Newfoundland, Little Port Harmon in western Newfoundland). We found there was no significant difference in elapsed entry attempt time between regions ($\beta1 = 0.231$, S.E. = 0.260, $t = 0.889$, $p = 0.374$), and we found no significant interaction between entry attempt outcome and region ($\beta1 = 0.046$, S.E. = 0.159, $t = 0.287$, $p = 0.774$). From our video data, the average elapsed time for successful entry attempts in the west was 163.7 s (SD = 169.5), and 120.5 s (SD = 124.1) in the southeast. The average elapsed time for failed entry attempts in the west was 129.8 s (SD = 195.4), and 122.5 s (SD = 204.8) in the southeast. Through our chi-squared test,

we failed to reject the null hypothesis that there was no association between the frequency of entry attempt outcomes and region ($\chi^2 = 0.558$, $df = 1$, $p = 0.455$). Based on all entry attempts within each region, the proportion of successful entry attempts was 14% in the west, and 16% in the southeast.

## DISCUSSION

### Video quality

In this study, we found underwater video to be an effective means of evaluating the Fukui trap as it actively fishes for invasive green crabs *in situ*, providing information that could not be inferred from catch data alone. However, there are inherent challenges associated with the collection of data from video recordings.

First, the illumination during overnight deployments was dim around the periphery of the FOV and the use of red lights had an impact on image quality due to high absorption of this frequency in water (*Williams et al., 2014*). Therefore, the number of approaches recorded during these deployments may have been less accurate than daytime deployments. This is a common issue when recording video in low-light environments (*Underwood, Winger & Legge, 2012*; *Favaro, Duff & Côté, 2014*). Both entry tunnels and the entry slits were clearly illuminated during overnight deployments. Therefore, the accuracy of entry attempt data remained consistent across all deployments. Second, we were limited to videos collected in June and July due to poor visibility caused by increased water temperature in mid-August. The videos collected in Fair Haven in August 2015 had to be disqualified due to excessive turbidity and suspended particulate material. Finally, as green crabs accumulated inside the Fukui trap, it became more difficult to track individual crabs as they made entry attempts. As the density inside the trap increased, our line-of-sight was often obstructed by green crabs already inside the trap. This may have had an effect on the number of entry attempts recorded in videos with high green crab densities, which could have ultimately influenced our calculations of entry attempt proportions.

### Evaluation of the six-step capture process

Through our video analysis, we have gained considerable insight into the performance of the standard Fukui trap as a tool for green crab mitigation, as well as the behaviour of the green crab in relation to the trap itself, other species, and other green crabs. These findings can be summarized using the framework of the six-step capture process (Fig. 1).

#### Step 1—Green crabs must be present in the ecosystem

The number of green crabs present in the areas where we deployed Fukui traps varied. Effective trapping requires that green crabs be present in sufficient numbers within the area being fished. Despite anecdotal evidence of established green crab populations at all sites sampled in 2015, most of our green crab catch was limited to either Fair Haven and Little Port Harmon (Table 1). We hypothesize that the low catch rates at the other locations could be attributed to environmental factors. Newfoundland experienced a prolonged winter in 2014–2015 with above normal ice extent, followed by a late spring warming (*DFO, 2016*). It has been shown that unusually low winter temperatures can result in mass

mortality of adult green crabs, and poor recruitment (*Crisp, 1964*; *Welch, 1968*; *Berrill, 1982*; *Beukema, 1991*). These low temperatures could have had an impact on green crab populations, producing less catch in certain areas than was seen in previous years (*Welch, 1968*; *Yamada & Kosro, 2010*).

When deploying the camera apparatus, we had to ensure that the camera would not breach the water's surface with the changing tides. To account for this, we deployed the cameras approximately 1 m below low tide depth. Green crabs are most commonly found in depths ranging from high tide levels to 5–6 m, and have been reported at depths of up to 60 m (*Crothers, 1968*; *Klassen & Locke, 2007*). Despite the minimum depth limitation dictated by the height of the camera above the Fukui trap, we are confident that the placement of our traps was sufficient to catch green crabs if they were present at each trapping location.

Bycatch at each location was generally low, particularly in areas where large numbers of green crabs were present (Table 2). This suggests that the Fukui trap has a minimal impact on native species, and is an appropriate trap for targeting green crabs in areas where other species are present. Presumed predation by green crabs causing bycatch mortality was rare, and limited to soft-bodied species such as winter flounder (*Pseudopleuronectes americanus*), cunner (*Tautogolabrus adspersus*), and sculpin (*Myoxocephalus sp.*) in Fukui traps containing large quantities of green crabs. We saw no mortality of rock crabs (*Cancer irroratus*) or American eel (*Anguilla rostrata*), and all living bycatch present in the Fukui trap upon retrieval was released alive.

### Step 2—Green crabs must detect the trap

Green crabs primarily use chemoreception to locate a food source (*Shelton & Mackie, 1971*). It did not take long for green crabs to locate and approach our baited Fukui traps after they settled on the seafloor. On average, the first green crab would approach the Fukui trap within four min. Therefore, if green crabs were present in the area where the trap was deployed, then the olfactory cues from the herring functioned as effective bait.

In our study, we did not examine the effects of water direction. However, other experiments on crustaceans have demonstrated that aligning a trap's entrances with the current can improve catch by leading the target species into the trap as they follow the bait plume (*Miller, 1978*; *Vazquez Archdale et al., 2003*). For this reason, when targeting green crabs with the Fukui trap it may benefit catchability to align the entrance tunnels with the water direction, so that crabs can follow the bait's odour trail directly into the trap.

### Step 3—Green crabs must approach the trap

We observed a range of different behaviours associated with green crabs approaching the Fukui trap. Some green crabs would make an entry attempt right away, entering the camera's FOV and proceeding directly to the entrance tunnel. In other instances, green crabs would move around the trap for long periods of time before discovering the entrance tunnel, or beginning an entry attempt. We frequently observed agonistic behaviour on and around the Fukui trap, especially once green crabs began to accumulate in the area. Green crabs would often cluster on top of the trap, situating themselves above the bait container (hanging inside the centre of the trap) as if they were guarding a food
source, a behaviour that has been noted with Dungeness crab (*Metacarcinus magister*) (*Barber & Cobb, 2009*). This behaviour would result in confrontations between green crabs as they fought to either defend their position, or to displace the green crab guarding the bait. It was common to witness one green crab pursuing another around the trap, or to observe one crab grasping and immobilizing another. Size did not appear related to which green crab was the aggressor. Green crabs not only exhibited intraspecific agonistic behaviours, but often engaged with other species near the trap. It was not uncommon for green crabs to display aggressive behaviour towards a larger fish species, such as winter flounder.

Because we could not individually identify crabs as they entered and re-entered the FOV, the number of approaches by green crabs to the Fukui trap does not represent the absolute number of individual crabs that approached the trap. This is a common challenge associated *in situ* camera studies (*Favaro, Duff & Côté, 2014*). Despite this caveat, every entry attempt we observed can be considered a unique event, regardless of whether a green crab approached multiple times. If a target species repeatedly approaches a piece of fishing gear, yet fails to be captured, this suggests a fundamental problem with the fishing gear itself that must be addressed. Furthermore, *Miller (1978)* demonstrated that unless a trap is efficient at capturing crabs shortly after they approach a trap, they will begin to accumulate around the trap. This will increase the frequency of agonistic interactions, causing many crabs to flee from the trap, reducing the capture efficiency. Therefore, for a trap to maximize efficiency, it must successfully capture a target species shortly after it approaches.

### Step 4—Green crabs must make an entry attempt

A total of 1,226 green crabs made entry attempts, of which the majority were unsuccessful ($n = 1,045$). We repeatedly observed four scenarios that resulted in failed entry attempts (i.e., AGON, PE, FE, DCE). These reoccurring failure scenarios occurred across all eight videos, demonstrating that both green crab behaviour, and Fukui trap performance issues remained consistent, regardless of location.

The least common reason for failed entry attempts was intraspecific and interspecific agonistic behaviour, which deterred or prevented green crabs from entering the Fukui trap. Aggressive behaviour is common in invasive species, allowing them to dominate over native species (*Rehage & Sih, 2004*; *Pintor et al., 2008*; *Weis, 2010*). The green crab is no exception, and is known for exhibiting both intraspecific and interspecific agonistic behaviour (*Rossong et al., 2006*; *Klassen & Locke, 2007*; *Souza et al., 2011*). This agonistic behaviour between green crabs has been shown to deter entry into baited traps (*Crothers, 1968*; *Gillespie et al., 2015*). Similar behaviour has been documented in red rock crab (*Cancer productus*), Dungeness crab (*Metacarcinus magister*), and American lobster (*Homarus americanus*) where they have been observed guarding the entrances to traps, or using their bodies to prevent other individuals from entering the trap (*Miller, 1978*; *Jury et al., 2001*; *Barber & Cobb, 2009*). We also observed this behaviour; however, these events only comprised 4% of all failed attempts, suggesting that it has a minimal impact on overall catchability.

Partial and full entry attempts occurred when green crabs gained access to the entrance tunnels, but did not make an active effort to pass through the trap entry slits. There was

little empirical evidence to explain these attempts beyond physical interactions between green crabs and the Fukui trap. The pereopods of green crabs could easily pass through the mesh of the Fukui trap, which would often cause them to become entangled or reoriented during entry attempts. Furthermore, the entry tunnels of a Fukui trap narrow towards the entrance slit. This limited the mobility of green crabs, and increased the likelihood that their pereopods would become entangled as they advanced further inside the entrance tunnel. This influenced the direction and orientation of the crab, and made it less likely that they would discover the entry slit in order to gain access to the inside of the trap.

The most common failure scenario occurred when a green crab had difficulty completing the entry attempt. This was characterized by the green crabs experiencing varying degrees of difficulty passing through the entry slit of the trap, and subsequently abandoning the attempt. The Fukui trap is designed so that a crab must force themselves through the entry slit, which remains tightly closed in its default position. However, even the most determined green crabs were often unable to enter the Fukui trap through these entry slits. A combination of mesh size and the restrictive opening of the trap entry slit made successful entries difficult. These same issues have been documented in a similar study with traps meant to target the Japanese rock crab (*Charybdis japonica*) (*Vazquez Archdale et al., 2003*). In this study, Archdale et al. observed crabs becoming entangled in the trap's netting material by their chelipeds and the spines on their carapace. They observed that forward-facing entry attempts would frequently result in entanglement, and difficulty in entering the trap. Furthermore, they observed that the trap's tight, narrow entry slits prevented crabs from squeezing in, forcing them to abandon entry attempts. For green crabs attempting to enter the Fukui trap, the predominance of DCE events suggests that the low catch rates are largely influenced by issues with the trap design itself, and not the behaviour of green crabs.

### Step 5—Green crabs must successfully enter the trap
Across all videos, we witnessed at total of 181 green crabs successfully enter the Fukui trap, suggesting that the capture efficiency of the trap is low. Successful green crabs were perseverant, often struggling for long periods of time before maneuvering themselves through the restrictive trap entry slit. Certain entry strategies appeared to assist green crabs in successfully entering the Fukui trap, and body orientation was an important factor in facilitating successful entries. Most successful entries occurred when green crabs approached the entry slit sideways or backwards. In doing so, they were less prone to becoming entangled in the mesh as they pushed their way into the trap. These same entry strategies have been documented in Japanese rock crabs attempting to enter baited traps (*Archdale, Kariyazono & Añasco, 2006*).

The force required to gain access to the Fukui trap would also make it challenging for green crabs to successfully enter the trap. Surprisingly, the bait container located in the centre of the trap would occasionally assist green crabs in making successful entry attempts. The tension of the trap entry slit made it difficult for green crabs to push themselves through; however, if they were able to make it partially inside the trap, grasping the bait container

would often allow them to pull themselves the rest of the way. This suggests that Fukui trap design would benefit from a proprietary mechanism to assist green crabs in pulling themselves into the trap.

There was great variability in success rates between videos, ranging from only 1% up to 33% (Table 3). When compared to similar studies of baited traps, the proportion of successful entry attempts into the Fukui trap is low. For example, traps used to capture Atlantic cod (*Gadus morhua*) and spot prawns (*Pandalus platyceros*) have successful entry attempt proportions of 22% and 46%, respectively (*Favaro, Duff & Côté, 2014*; *Meintzer, Walsh & Favaro, 2017*).

Furthermore, there was a disconnect between attraction to the trap and final catch. The number of approaches was positively correlated with entry attempts, demonstrating that if there were many approaches to the trap, there were generally many entry attempts (Figs. 6A and 6B). The large number of attempts seen in Fig. 6B indicates that green crabs were actively trying to enter the Fukui trap. However, Fig. 6C shows that this does not necessarily reflect how many green crabs were actually captured.

Figure 6C demonstrates that catch is not an accurate representation of entry attempt effort, as the success rate varied widely. Certain videos (e.g., PH5) had many green crab entry attempts, resulting in comparatively high catch. However, some videos (e.g., FH3, PH1) had many approaches and attempts, yet caught very few crabs. We hypothesize that the varying success rates may have been due to the condition of the specific Fukui trap used. For example, if the metal frame of the trap was distorted in such a way that the tension of the entry slit was altered, this could affect how well a green crab is able to enter the trap. Alternatively, if the mesh of the trap is worn or sagging, this could promote successful entries by making the entry slit less restrictive. Although we did not record the condition of the Fukui traps used in our study, future experiments should test the performance of specific traps as a factor that could influence catch. This hypothesis emphasizes the importance of regularly inspecting the condition of the Fukui trap in order to promote successful entry attempts.

The variable success rates not only suggest there may be a fundamental problem with the design of the Fukui trap, but that final catch does not necessarily reflect the abundance of green crabs in the vicinity of the trap at the time of deployment. Over the course of a deployment, many green crabs may attempt to enter a trap. However, as we have shown in this study, this effort is not necessarily reflected in the number of crabs that are captured. This suggests that final catch could produce a biased perception of low green crab abundance in the area being fished. Other studies of crustacean catchability have demonstrated that traps can lead to biased estimates of CPUE and abundance (*Murray & Seed, 2010*; *Kersey Sturdivant & Clark, 2011*; *Watson & Jury, 2013*). For green crabs, local abundance is often estimated by catch rate (*Gillespie et al., 2007*; *Duncombe & Therriault, 2017*). From an invasive species management perspective, this shows that there may be more green crabs in an area than is suggested by catch data alone, emphasising the importance of not relying exclusively on catch data to estimate green crab populations in invaded areas.

### Step 6—Green crab must not exit the trap

Over the 73 h of video we analyzed, we did not observe a single escape from the Fukui trap, demonstrating that although it is difficult to enter the trap, once inside there is very little chance of a green crab escaping. However, it should be noted that we were not always able to retrieve the trap before the end of the video recording. Therefore, our final catch numbers do not necessarily correspond to what was observed in the video. Given the low rate of successful entry, the benefits of a highly secure trap that prevents escapes are lost when compared to the potential number of green crabs that could be captured if the entrance to the trap was less restrictive to begin with. To be more efficient, the Fukui trap needs to have a balance between effective catch and the risk of potential escapes.

## Regional performance of the Fukui trap

The green crab is considered a global invader, and has established populations on almost every continent around the globe (*Yamada, 2001*; *Carlton & Cohen, 2003*). The expansive distribution of invasive green crab populations in North America alone, coupled with variations in genetic origin, suggests that there may not be a one-size-fits-all approach when responding to green crab invasions. That being said, the Fukui trap is being used on both the east (*Matheson & Gagnon, 2012*; *Rossong et al., 2012*; *McNive, Quijon & Mitchell, 2013*; *Best, McKenzie & Couturier, 2014*) and west (*Yamada et al., 2005*; *Yamada et al., 2008*; *Jensen, McDonald & Armstrong, 2007*; *Duncombe & Therriault, 2017*) coasts of North America, and remains the trap of choice for green crab mitigation due to its relative effectiveness, durability, and ease-of-use compared with other traps (CH McKenzie, pers. comm., 2015).

In Newfoundland, we anticipated that genetic differences in aggression and foraging behaviour might influence how green crabs interacted with the Fukui trap. However, we saw little variation in the performance of the Fukui trap from one study site to the next, and there was no statistically significant differences in trap efficiency between regions. This suggests that the factors that contribute to high entry attempt failure, and therefore limit catch efficiency, are underlying problems with the Fukui trap itself and are not influenced by behavioural variations in local green crab populations. If these underlying factors that limit catch efficiency can be addressed and corrected, then we expect that catch efficiency can be improved wherever Fukui traps are being utilized as a mitigation tool, regardless of genetic differences and regional green crab characteristics.

## Efficiency and modification

Only 16.0% of green crab entry attempts were successful, demonstrating that there is much room for improvement in the performance and efficiency of the Fukui trap. Still, the Fukui trap is a common choice for green crab mitigation across Canada, and intensive trapping has proven to be an effective technique for reducing green crab populations (*Gillespie et al., 2007*; *DFO, 2011a*; *DFO, 2011b*). It has been shown that continuous trapping can cause a demographic shift towards a younger population, with reduced body mass and reproductive potential (*Duncombe & Therriault, 2017*). Furthermore, continuous trapping can gradually reduce the average carapace width of green crabs in an invaded area by

removing larger individuals from the population (*Duncombe & Therriault, 2017*). This size decrease causes a shift in the ecological role of green crabs from primary predators, to potential prey for native shorebirds and crustaceans (*DFO, 2011a*). Furthermore, it has been shown that in areas where intensive trapping has occurred that the abundance of native species increases over time (e.g., rock crabs) (*DFO, 2011a*). Therefore, despite the limitations of the Fukui trap, it remains an important tool for reducing green crab populations in invaded ecosystems.

Based on our video observations, we believe there is scope to develop an improved Fukui trap that will facilitate the entry of green crabs into the trap. The problems associated with the design of the Fukui trap are predominately mechanical issues, and can likely be addressed through modifications. We propose three simple modifications that would likely improve the efficiency of the Fukui trap: First, the entry slit of the trap needs to be expanded slightly to allow green crabs to pass through more freely. This could be accomplished using string, cable ties etc. to secure the entry slits in a partially-opened position. Second, the Fukui trap entrance tunnels could be constructed using a smaller mesh that would prevent green crabs from becoming entangled or snagged during entry attempts. This could quickly be accomplished by overlaying the existing mesh with a finer mesh, and securing it in place. Finally, green crabs would benefit from a fixed object on the inside of the Fukui trap that they could grasp in order to assist in pulling their bodies through the entry slit. This could be accomplished by simply affixing a piece of string, wire, mesh etc. across the width of the trap, on the interior side of the entrance slit. In addition to these design modifications, future studies could also test whether the addition of artificial lighting to the Fukui trap can be used to improve green crab CPUE, as seen with snow crab (*Chionoecetes opilio*) traps (*Nguyen et al., 2017*).

Any modifications to the Fukui trap would have to be tested to quantify the trade-offs of an altered trap design. If modifications were to alter the restrictive entry slit, this could increase the proportion of successful green crab entry attempts, but it could also impact retention (i.e., cause an increase in green crab exits). Furthermore, it is possible that a Fukui trap modified to capture more green crabs, may also catch more bycatch. However, if these design modifications are found to be effective, and the negative trade-offs are minimal, then this will greatly increase the number of green crabs that are removed from invaded ecosystems during mitigation efforts. Additionally, a more efficient Fukui trap will mean higher CPUE, maximizing trap usage for mitigation and control. Ultimately, a more efficient Fukui trap will help to control green crab populations in order to preserve the function and integrity of ecosystems invaded by the green crab.

## CONCLUSIONS

Our study represents the first formal investigation into the performance of the Fukui trap as a mitigation tool for the invasive green crab in Newfoundland. Our use of underwater video was a novel approach that allowed us to accurately determine the capture efficiency of these traps in a way that would be unachievable from catch data alone. Through the use of underwater video, we were able to gain insight into the efficiency of the Fukui trap, as
well as the interactions that occur around and inside these traps as they are actively fished for green crab *in-situ*. Although our results revealed the rate of successful entries into the Fukui trap was low, we are confident that the mechanical inefficiencies of the trap can be addressed through simple modifications that will increase their CPUE. Furthermore, we were able to conclude that the underlying mechanisms contributing to low capture efficiency remained consistent regardless of the region or the local green crab population. The versatility of the Fukui trap as a control method for green crabs has contributed to its widespread use on both the east and west coast of Canada. Therefore, if the performance and efficiency of the Fukui trap can be improved then this will benefit green crab mitigation efforts wherever these traps are being used.

## ACKNOWLEDGEMENTS

We thank many individuals for their assistance and contributions to this project. We thank DFO for providing the Fukui traps and bait used in this study. We thank staff at the Marine Institute's Centre for Sustainable Aquatic Resources (Terry Bungay and George Legge) for assistance in constructing and testing the camera apparatus. We acknowledge DFO staff from the Aquatic Invasive Species program and the Ecological Sciences Section (Kyle Matheson, Ashley Bungay, Haley Lambert, Dave Forsey, Rebecca Raymond, and Bob Whalen) for their assistance with fieldwork, as well as DFO Fisheries Patrol Officers (Sherry Pittman and Kim Sheehan) for assistance with fieldwork in the Bay of Islands, NL. We also acknowledge the MUN Field Services team (Andrew Perry, Zach Ryan, and George Bishop) for their assistance in the field. We thank Sheldon Peddle with ACAP Humber Arm for providing us with additional bait, and for delivering replacement camera equipment while in the field. We thank Bob Hooper with the MUN Bonne Bay Marine Station for providing access to a boat, field resources, and accommodation. For fieldwork conducted in Fox Harbour we thank Gerard O'Leary for access to private property for the deployment of our camera equipment. Finally, we thank one anonymous reviewer, Leslie Roberson, and our academic editor, Dr. Donald Kramer for their constructive reviews of our manuscript, which greatly enhanced the final paper.

### Funding

This project was funded by a Marine Environmental Observation Prediction and Response (MEOPAR) Early-Career Faculty Development Grant awarded to Brett Favaro (EC1-BF-MUN). Jonathan A. Bergshoeff and Nicola Zargarpour were supported by Ocean Industry Student Research Awards from the Research and Development Corporation of Newfoundland and Labrador (5404-1915-101 and 5404-1914-101, respectively). Funding was also provided by the Canadian Centre for Fisheries Innovation (H-2015-06), and the Newfoundland and Labrador Department of Fisheries and Aquaculture (currently, Department of Fisheries and Land Resources) Fisheries Development and Diversification Fund (NH-77836). The funders had no role in study design, data collection and analysis, decision to publish, or preparation of the manuscript.

## Grant Disclosures

The following grant information was disclosed by the authors:

Marine Environmental Observation Prediction and Response (MEOPAR): EC1-BF-MUN.
Research and Development Corporation of Newfoundland and Labrador: 5404-1915-101, 5404-1914-101.
Canadian Centre for Fisheries Innovation: H-2015-06.
Newfoundland and Labrador Department of Fisheries and Aquaculture: NH-77836.

## Competing Interests

The authors declare there are no competing interests.

## Author Contributions

- Jonathan A. Bergshoeff conceived and designed the experiments, performed the experiments, analyzed the data, wrote the paper, prepared figures and/or tables, reviewed drafts of the paper.
- Cynthia H. McKenzie conceived and designed the experiments, contributed reagents/materials/analysis tools, reviewed drafts of the paper, additional field support and resources from Fisheries and Oceans Canada.
- Kiley Best conceived and designed the experiments, performed the experiments, reviewed drafts of the paper.
- Nicola Zargarpour performed the experiments, reviewed drafts of the paper.
- Brett Favaro conceived and designed the experiments, contributed reagents/materials/analysis tools, reviewed drafts of the paper, supervision and advice throughout the study.

## Field Study Permissions

The following information was supplied relating to field study approvals (i.e., approving body and any reference numbers):

This project was reviewed and approved as a 'Category A' study by the Institutional Animal Care Committee at Memorial University (Project # 15-02-BF). All field research was conducted under experimental licenses NL-3133-15 and NL-3271-16 issued by Fisheries and Oceans Canada.

## Data Availability

The raw data is provided in the Supplemental Files.

## Supplemental Information

Supplemental information for this article can be found online at http://dx.doi.org/10.7717/peerj.4223#supplemental-information.

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
