# Peer review of "Using underwater video to evaluate the performance of the Fukui trap as a mitigation tool for the invasive European green crab (Carcinus maenas) in Newfoundland, Canada"

_PeerJ, doi:10.7717/peerj.4223_

## Round 0.1 · original submission · Minor Revisions

Both reviewers found your study important, well analyzed, and generally well presented and recommended acceptance after minor revisions. I concur with this view, but have quite a few additional suggestions for improving the clarity and focus of the presentation. In particular, I found that the Discussion was rather wordy and redundant and that some sections did not seem to be arranged logically.

I have made minor grammatical suggestions using highlights to indicate problem areas and inserted comments to indicate potential solutions. To facilitate your revision, I used the pdf already analyzed by Reviewer 2; you can recognize my comments by my initials (DLK). Reviewer 2 sometimes used the cross-out function but I only highlighted sections. In general, I agreed with the reviewer's comments so did not repeat them. You may treat my comments below as if they were a third review, i.e. make appropriate changes if they are valid and provide a careful rebuttal if they are not. You don't need to respond to my comments made directly on the manuscript unless you disagree. For the reviewer's comments directly on the manuscript, explicitly mention in your rebuttal only any that were not included in the 'comments for the author' part of the review as well as any grammatical suggestions with which you disagree.

Editor's Comments

Keywords: I think that Behaviour, Behavioural Ecology and Marine Biology are too general for keywords; they describe the broad subjects of the manuscript and would not be used for a search. However, you might consider adding the family name of the green crab and alternative terms for traps and trap effectiveness that might be used by researchers.

Reviewer 2 questioned the apparent contradiction in the process of selecting which deployments to analyze. I also stumbled here and had to re-read to recognize that you had different criteria for inclusion of samples when you analyzed the effect of the camera on catch and when you carried out the video analysis of trap entry. Even though the subsection headers are clear, you might consider revising the wording in the video section to make it a bit more explicit that different criteria were required for the two analyses.

The manuscript is not consistent in presenting means and variation. Choose one format and use it consistently, as suggested by reviewer 2. Also, if you indicate in the methods which measure of variability (SE, SD, CL) you are using, you can avoid repetition. I believe it is widely accepted that SD rather than SE is the appropriate measure for descriptive data. I am glad that you have chosen to often include minimum and maximum values, as these appear quite relevant in many cases.

L300. I don't think you need to include Fig. 8 as it does not show very much. If you want to make the point concerning lack of correlation between total entry attempts and percent success among trials, you could run and simple correlation and provide the outcome. However, I don't see what sort of relationship you would expect. Are you thinking that trials if a trap was harder to enter the crabs might make more attempts? If this is a useful relationship to examine, explain to the reader why it might be important. I note that you refer to this figure to indicate variation in entry per attempt, but I think that there are more effective ways to indicate this variation.

Fig. 9 caption should include the format of the boxplot (i.e. what the limits of the box, whiskers, dots, etc. represent).

L305ff. The text is redundant to Fig. 9 in repeating the duration of different types of entry failure. I suggest removing the times from the text.

L317ff. I don't think that Fig. 10 adds much useful information. With information about the statistics as suggested by the reviewer, I think this paragraph is sufficient without the figure.

L324-326. Fig. 11 should be either restructured or removed. It would be extremely difficult to detect a site effect from the stacked bars. You use the figure only to support one (or two) brief statements. It is not clear if this is your evidence for a lack of saturation effect also. However, I question whether you could detect saturation from this sort of data. How can saturation be distinguished from a local depletion effect? Perhaps by a decrease in entries independent of a decrease in approaches? If you want to discuss whether the capture rate is a linear or decreasing function of density, the best way is to test alternative functions with AIC, including consideration that the total numbers are not independent within a trap, so you have to compare arrivals or entries to total density. As an aside, the tendency for the largest proportion of successful entries to occur at intermediate values would be useful to test in the future. I seem to recall that in some fish traps studies, there is an attraction component at low densities. If there was an interaction between an increasing and decreasing effect of density in the trap, it would be a larger challenge to detect with simple linear statistics.

L359. This statement appears to be incorrect. Table 1 includes no information about numbers in the environment and your study did not assess this directly. The rest of the paragraph arguing that low catch rates might relate to low abundance is ok.

L376-378. 'Environmental factors' is too vague. There are environmental factors that influence abundance and environmental factors that might influence CPUE (e.g., factors that affect mobility or hunger of the crabs present or the olfactory plume from the bait). This paragraph could end on L376.

L388ff. With regard to detection, do you have any comments on the effect of currents? Did the crabs approach mainly from one direction (down current), for example?

L393 and elsewhere. In the Discussion, you should not repeat data details that are (or should be) in the Results. Only focus on the relevant information, which in this case might be the that the 'first crab approached within about one to ten minutes' or that 'the first crab approached within 4 min, on average'.

L394. If you are going to make this point, it might be worth stating in the Methods whether the herring was fresh or previously frozen, whole or cut up.

L401-402. 'Survey' and 'deciding' imply that you are aware of the functional significance and neural processes of the crabs. I suggest replacing by more empirical behavioural terms such as 'moving around the traps', 'pausing near the trap', or whatever you mean by 'surveying' and 'starting to enter'.

L418. Not clear what you mean by 'absolute'

L419-421. This sentence seems more relevant to Step 5 than Step 3.

L423ff. The sections on entry attempts and successful entry need revision. You don't have to repeat details of quantitative results, and the descriptions of context of failures (e.g., L441-449, L451-457, 466-474) to enters are basically results. You should move these to results and focus this section on a synthesis of main reasons for failure and relation to previous literature. L475-494 includes a mixture of results (e.g. variability among trials) and discussion. Please go through these paragraphs and allocate the actual observations to the Results and focus the Discussion on interpretation (e.g., variation in trap conditions).

In addition, I question the allocation of material to the different steps. The section on entry attempts (Step 4) seems to focus on failure of successful entry which is the subsequent step. For example, the grasping of objects to assist entry is mentioned in both sections, dividing discussion of a single concept.

L485ff. If you had evidence that certain individual traps had consistently different success rates than others, this would strengthen your argument here.

L493. What do you mean by a variable influence?

L495-500. I think your logic is not fully developed here. You don't have direct data on the number of crabs in the area, so the relationship you observed is between approaches and captures. I agree that highly variable capture rate will cause variation in the estimate of abundance, but there are usually many more individuals in the 'ecosystem' (is this really term you want for the local area?) than captured in a trap. Variation in each stage of capture will cause variation in estimates. However, variation is a normal part of ecological studies, often countered by larger sample sizes, and it is not clear that you have you found more variation than other studies of this or other species. You need references to show that other researchers use catch data to estimate population size.

L514-537. These paragraphs are too long and wordy and not directly relevant. Condense and add relevant parts to Introduction with only a brief reference here in the context of lack of differences.

Given that you did not find any regional differences but had relatively low replication, the rest of this section should be greatly condensed with a focus on interpretation rather than background and results. Is there any relevant literature showing that genetic or other regional differences result in substantial changes in the components of capture success?

L574. The logic regarding reduction in carapace size and susceptibility to predators is not well developed. Presumably, substantial trapping removes larger individuals from the population leaving smaller individuals that were already there. These individuals are not obviously more susceptible to predators as a result of the removal of larger conspecifics.

L577-584. My concerns regarding your discussion of the abundance:capture relationship mentioned above also apply here. Variability per se is not a problem as long as sample sizes are adequate and the confidence interval respected.

L585ff. The next paragraph is off topic as your study does not contribute directly to CPUE as a measure of abundance. If needed, you need references to support sources of variation in capture (L587).

L594-598 is redundant. Incorporate L598-602 with previous discussions of capture success.

L603-616. These paragraphs are very wordy and redundant. The important thing is that you indicate potential improvements. The implications of an improved trap are obvious and this manuscript does not develop improvements. A more explicit linking of suggested improvements to apparent obstacles to entry would be helpful.

L587. References needed for this statement.

Wording: You use green crab as a collective noun, for example, the 'number of green crab' rather than the 'number of green crabs'. I recognize that this is an acceptable usage and common in fisheries and won't insist on a change. Nevertheless, I think that using the plural crabs would be clearer to many readers.

Grammatical issues. There are several repeated grammatical errors in your manuscript. It might be useful to briefly review the principles involved as many useful online guides exist. You should use commas to separate independent clauses and dependent clauses that start a sentence. Try to avoid run-on sentences: two separate sentences joined by a conjunctive adverb such as therefore and however. Use a period or semicolon to divide the two independent clauses and a comma after the conjunctive adverb.

Reviewer 1 ·

Basic reporting

This article, addressing the trapping efficiency of the Fukuii fish trap, is a very important study that should be of great interest to those biologists who use this trap to monitor and control the European green crab. Green crab researchers in Australia and on both coasts of North America are using this trap because it is light and can be folded. However, no one has ever questioned its trapping efficiency. Knowing that the efficiency is only 16% is especially important to programs designed to “eradicate” newly discovered satellite populations. It is important to note that poor catches may not mean that a population has been reduced to a low enough level to prevent self-seeding.
The article is well written and easy to follow. The introduction includes all the relevant references and clearly states the context of this study.
The figures are clear and well described. I wonder if Figure 6, showing catches with and without the camera can be left out. All that would be needed is a statement that there was no significant difference in the catches.

Experimental design

The experimental design is rigorous and well described. The fact that the authors used of a no-camera control treatment and deployed the traps at various sites and various conditions is very impressive.

Validity of the findings

The data are robust and the analysis is sound. From this rigorous study we learned that most meaningful data can be obtained at sites with high crab densities, during daylight and with good visibility. Knowing these parameters will make it easier to test modifications to the Fukuii trap that will increase catch rate. The number 1 modification for increasing catch rate would be to work on “difficulty in completing entry”. I am suggesting that the authors describe some modifications that could be tested to achieve this goal. Even though the authors did not test any of these modification, they are in the best position to advise fellow green crab biologists for getting better catch rates.

Additional comments

This is a very needed piece of information for all researchers who use the Fukuii trap. Knowing the traps limitation needs to be incorporated in any monitoring and control studies. It would help if the authors made some suggestions for improving the design of the Fukuii traps. Researchers using new traps with tight fitting openings may not trap as many crabs as researchers using older traps with sagging mesh openings. That could lead to a difference in CPUE.

·

Basic reporting

Writing is generally clear and easy to follow, but some sentences should be more concise to improve flow (specific changes annotated in PDF). The discussion could be shortened by removing some redundant sentences. Some important references are missing (annotated in PDF).

Experimental design

This study addresses an important gap in our understanding of the efficacy of a widely-used tool.
The authors do a good job of clearly specifying their objectives, which are to 1. Evaluate performance and efficiency in terms of catchability (CPUE of Fukui traps) and 2. Understand patterns in catchability between different sites. To this end, they are only evaluating steps 3-6 of catching a crab in a Fukui trap: the approach, locating the entrance/attempting entry, and entering the trap. Importantly, they are not evaluating catch relative to abundance of crabs around the trap, nor are they considering the retention of target catch until the gear is hauled.
There is some confusion about the Methods: which videos were included in the analysis. The authors use an “elimination threshold” of a mean of 10 green crabs caught per deployment (Line 170) because “effective trapping requires that green crab be present in sufficient numbers within the area being fished” (line 359). But, in section 2.4.1 the authors say that videos were eliminated based on no or low “activity,” determined by counting the approximate number of green crab present in the FOV (Line 186). Please clarify whether videos were eliminated based on low catch or on low abundance around the trap, so that the text matches with SI Table 2.

Validity of the findings

The dataset is small in terms of number of videos (8 out of 37 deployments used in analysis) but the study is focused only on the approach, attempted entry, and entrance into the trap. To that end, there were 2,373 approaches, which is a sufficiently large dataset to make conclusions about the performance of the Fukui trap.
Conclusions are well-stated, and speculation regarding the causes of the results are clearly stated as such. However, the potential importance of additional factors should be mentioned (e.g. sex and year, annotated in PDF) as well as the day-night effect, which is interesting and not given enough attention in the discussion. Be careful to specify where you are referring to the effect of the camera on the trap, versus environmental factors or the trap itself (annotated in PDF).

Additional comments

This is a good study that is beneficial to the literature and applicable to management of invasive green crab. The manuscript structure is easy to follow (the 6 steps are clearly described).
I have made the following suggestions to improve the validity of the findings and the readability of the manuscript: (also annotated in yellow comment boxes on PDF)
1. Line 117: Add more recent reference. Have any studies specifically been done on C. maenas and lights? See annotation
2. Line 122: Citing Kahle & Wickham 2013 doesn’t make sense unless you add that you made figures with ggplot2
3. Line 127: Should mention habitat type as this could have an effect on crab abundance, behavior, and therefore catchability (e.g. Bellchambers 2013, “Assessing the effectiveness of two methods of habitat characterisation for understanding species habitat relationships, using the western rock lobster (Panulirus cygnus George)”)
4. Line 130: reword suggested
5. Line 132: scientific name of herring first time you mention it
6. Line 170: Did you test the effects of day/night?
7. Line 189: Explain that this is the “elimination threshold.” See general comments about the confusion with the video selection criteria/elimination threshold
8. Line 238: state potential relevance of genetic differences for catchability
9. Line 272: Which sample subset are you referring to? Only used 8 videos in analysis?
10. Line 306: Present in order (most common – least common, or visa versa)
11. Line 318: Include model results (p values for study site effect)
12. Line 326: How is saturation defined in the literature?
13. Line 378: Inconsistent with your later speculation about trap construction.
14. Line 386: Clarify – all bycatch present in the trap upon retrieval were released alive
15. Line 390 – rephrase, “within seconds” is misleading
16. Line 451 and 454: Rephrase these sentences: “often” and “not uncommon” are misleading as this was the “least common reason for failure”
17. Line 463: suggested changing word order
18. Line 478: The disconnect is not necessarily between abundance and final catch, it’s between attraction to the trap and final catch.
19. Line 479: Suggested reword
20. Line 494: Also individual traps as a factor in future studies
21. Line 497: Should mention other studies with this same finding: Watson & Jury 2013 (“The relationship between American lobster catch, entry rate into traps and density”), Sturdivant & Clark 2011 (“An evaluation of the effects of blue crab (Callinectes sapidus) behavior on the efficacy of crab pots as a tool for estimating population abundance”)
22. Lines 515 and 518 belong in the introduction
23. Lines 526 – 535: most of this should be moved to the introduction
24. Line 545: What is CHM?
25. Line 585: Specify of C. maenas (see comment)
26. Line 586: Should also consider the effect of year (Tremblay et al. 2006) and sex – mention other relevant studies
27. Line 590: consider the peer-reviewed literature regarding the efficacy of sampling techniques (e.g. Stobart et al. 2015, "Performance of Baited Underwater Video: Does It Underestimate Abundance at High Population Densities?", Roberson et al. 2017, "Potential application of baited remote underwater video to survey abundance of west coast rock lobster Jasus lalandii").
28. Line 604: This sentence is redundant, consider removing
29. Line 609: Should also consider testing lights to improve CPUE (See comment for line 117 – Nguyen 2017).
30. Line 610: redundant sentence, consider removing.
31. Line 615: Should mention future studies on tradeoffs in trap design, also potential effects on bycatch rates
32. Figure 11: Hard to read, change red color scale

---

## Round 0.2 · accepted · Accept

The authors have carried out appropriate changes in response to the reviews. The manuscript can now be published. I have submitted a pdf with a small number of very minor corrections indicated by highlighting.